# The Bottlenecks of Preparing Virus Particles by Size Exclusion for Antibody Generation

**DOI:** 10.3390/ijms232112967

**Published:** 2022-10-26

**Authors:** Chi-Hsin Lee, Peng-Nien Huang, Pharaoh Fellow Mwale, Wei-Chu Wang, Sy-Jye Leu, Sung-Nien Tseng, Shin-Ru Shih, Liao-Chun Chiang, Yan-Chiao Mao, Bor-Yu Tsai, Nhlanhla Benedict Dlamini, Tien-Cuong Nguyen, Chen-Hsin Tsai, Yi-Yuan Yang

**Affiliations:** 1Ph.D. Program in Medical Biotechnology, College of Medical Science and Technology, Taipei Medical University, Taipei 110301, Taiwan; 2School of Medical Laboratory Science and Biotechnology, College of Medical Science and Technology, Taipei Medical University, Taipei 110301, Taiwan; 3Research Center for Emerging Viral Infections, Chang Gung University, Taoyuan 333423, Taiwan; 4Division of Infectious Diseases, Department of Pediatrics, Linkou Chang Gung Memorial Hospital, Taoyuan 333423, Taiwan; 5Graduate Institute of Medical Sciences, College of Medicine, Taipei Medical University, Taipei 110301, Taiwan; 6Department of Microbiology and Immunology, School of Medicine, College of Medicine, Taipei Medical University, Taipei 110301, Taiwan; 7Department of Laboratory Medicine, Linkou Chang Gung Memorial Hospital, Taoyuan 333423, Taiwan; 8College of Life Sciences, National Tsing Hua University, Hsinchu 300040, Taiwan; 9Division of Clinical Toxicology, Department of Emergency Medicine, Taichung Veterans General Hospital, Taichung 407219, Taiwan; 10Navi Bio-Therapeutics Inc., Taipei 10351, Taiwan; 11Department of Ophthalmology, Taipei Medical University Hospital, Taipei 110301, Taiwan; 12Core Laboratory of Antibody Generation and Research, Taipei Medical University, Taipei 110301, Taiwan

**Keywords:** Enterovirus 71 (EV71), hand-foot-mouth disease (HFMD), yolk-immunoglobulin, phage display technology, single-chain variable fragment (scFv)

## Abstract

Enterovirus 71 (EV71) is the major etiological agent contributing to the development of hand-foot-mouth disease (HFMD). There are not any global available vaccines or antibody drugs against EV71 released yet. In this study, we perform the virus immunization in a cost-effective and convenient approach by preparing virus particles from size exclusion and immunization of chicken. Polyclonal yolk-immunoglobulin (IgY) was simply purified from egg yolk and monoclonal single-chain variable fragments (scFv) were selected via phage display technology with two scFv libraries containing 6.0 × 10^6^ and 1.3 × 10^7^ transformants. Specific clones were enriched after 5 rounds of bio-panning and four identical genes were classified after the sequence analysis. Moreover, the higher mutation rates were revealed in the CDR regions, especially in the CDR3. IgY showed specific binding activities to both EV71-infected and *Coxsackievirus* 16-infected cell lysates and high infectivity inhibitory activity of EV71. However, while IgY detected a 37 kDa protein, the selected scFv seemingly detected higher size proteins which could be cell protein instead of EV71 proteins. Despite the highly effective chicken antibody generation, the purity of virus particles prepared by size exclusion is the limitation of this study, and further characterization should be carried out rigorously.

## 1. Introduction

Infection causes by virus such as enteroviruses (EVs) are a universal medical problem. EVs are considered the most prevalent viruses infecting humans worldwide. To date, there are at least 106 types of EVs known to infect human and dozens are commonly detected [1]. Symptoms caused by EVs infection vary from mild respiratory disease, and hand-foot-mouth disease (HFMD) to severe meningitis, myocarditis, and gastrointestinal disease, but most of the infections are asymptomatic [2,3,4]. EV71, which is belonged to *Picornaviridae* family, is one of the major serotypes of human enterovirus type A [5], and is considered to be the dominant etiological agent of HFMD, encephalitis, and central nervous system disorders in young children worldwide [6,7,8]. Epidemic outbreaks of EV71 with a 7% of high mortality rate have been reported in Europe [9,10]. Large-scale epidemics of EV71 have also been reported in Asia, including Taiwan, China, Singapore, Vietnam, and Cambodia, among the patients diagnosed with EV71 HFMD has resulted in the death of from 0.5 to 19% [11,12,13,14,15]. The infections associated with these diseases have posed a serious global public health threat [16]. Therefore, developing a cost-effective and more tolerable therapeutic strategy for the control and prevention of EV71 is favorable worldwide.

An EV71 icosahedral particle consists of 60 copies of four capsid proteins, (VP1 [viral protein 1], VP2, VP3, and VP4). The viral polyprotein is further cleaved during the virus assembly process to form VP0, VP1, and VP3. The VP2 and VP4 are then formed from the cleavage of the precursor VP0. This process is auto-catalyzed by viral RNA in mature virions [10]. These protein structures, present in both *enteroviruses* and *coxsackieviruses*, are considered to be immunogenic and potential targets for the development of vaccines or antiviral drugs. Several vaccines have been produced targeting EV71 and tested in clinical trials, however, it is approved only in China, and are not available globally [17,18]. Although vaccination is still an excellent approach to fight the diseases, vaccines are prone to reduce efficacy due to viral mutation. To avoid the problem associated with viral mutations, antibodies targeting conserved regions of viral proteins can be generated as an alternative. Antibodies have been widely developed and tested against different viruses, and studies have shown a significant neutralization activity against rabies, dengue, and *Coxsackievirus* A16 (CA16) [19,20,21]. Unfortunately, no antibody-based therapy has been approved by US FDA for the treatment of EV71 infection.

Antibody treatment has been an effective therapy for a broad range of conditions including cancers and viruses [22]. Chickens might be a substitute for mammals as antibody sources because they are inexpensive to foster and easy to handle [23]. In addition, the production of large amounts of polyclonal antibodies called immunoglobulin Y (IgY) from the yolk of chicken eggs is much easier for purification without bleeding [24]. Many studies have reported that the IgY antibodies, which does not react with certain mammalian complement system or Fc receptor, used in passive immunization treatments might be a cheaper alternative therapeutic strategy [25]. Chickens are also a cost-effective animal to generate specific monoclonal antibodies using phage display technology [26]. Phage display technology has been effective for over a decade, and has been widely used in to develop antibodies that could potentially target different diseases for detection and therapy [27]. A small antibody fragment, such as a single-chain variable fragment (scFv) or an antigen-binding fragment (Fab), is highly suitable for using phage display technology to generate. In addition, small antibody fragments nowadays have been emerging as new applications in diagnosis and therapy for the biotechnology market. Recently, numerous scFv antibodies have been expressed in *E. coli* and their binding activity has been tested against different viruses and targets [28,29,30,31]. Studies have also proved that passive immunization of neutralizing proteins could provide protection in mice from fatal EV71 infection in vivo [32].

In this study, we attempted to generate and evaluate chicken antibodies, including polyclonal IgY and monoclonal scFv antibodies, from the immunization of virus particles purified from size exclusion. Immunogenic proteins play the most important role in antibody generation. Because total amount and quality of virus particles purified by using sucrose gradient were not often satisfactory, we used the 100 kDa size exclusion column to prepare the EV71 virus particles instead of traditional sucrose gradient method. This approach is notably cost-effective and convenient, not only in animal immunization but also in the virus purification when compared to the traditional sucrose gradient method. After chicken immunization with EV71 virus particles, IgY antibodies were purified from egg-yolks and scFv antibodies were selected from libraries constructed by phage display technology. Antibodies were analyzed for their binding abilities on Western blot and ELISA assays, and for their neutralization abilities, which were evaluated using an in vitro neutralization assay. According to this simple approach, we are still looking forward to generating antibodies with high potential for detection and neutralization applications.

## 2. Results

### 2.1. Analysis of Chicken IgY Antibodies after Immunization

After virus infection to RD cells, the EV71 virus particles were purified and used for chicken immunization seven times. To assess the immune response in chicken, purified IgY antibodies from various stages of immunization were used for binding to EV71 virus particles (Figure 1A). The chicken did not lay eggs during the period of the 2nd and 4th immunization. The purified IgY from pre-immunization and the 1st immunization did not show binding activities to EV71 virus particles. In contrast, specific IgY antibodies were elicited after the 3rd immunization and showed a strong binding ability to EV71 virus particles, rather than to BSA at 5000× dilution. The titers might have reached a plateau after 3rd immunization. Moreover, when a serial dilution was performed, the IgY antibody from 7th immunization had also shown a strong binding activity to EV71 virus particles (OD > 1.0) at 16,000× dilution, moderate binding activities to EV71 virus particles (OD > 0.5) at 64,000× dilution (Figure 1B). These results demonstrated that the humoral response in chicken was successfully elicited by using EV71 virus particles for immunization.

### 2.2. Construction and Selection of Phage Displaying scFv Antibodies

In order to select scFv antibodies, the cDNA was first synthesized using total RNA, which was extracted from the chicken spleen, then was used for amplification of light chain variable regions (V_L_) and heavy chain variable regions (V_H_) genes. The V_L_ and V_H_ genes were linked by a short or a long peptide linker to form the full-length scFv genes by an overlapping PCR. The scFv genes were cloned into a phagemid vector for the construction of scFv antibody libraries, the short linker and long linker libraries containing 6.0 × 10^6^ and 1.3 × 10^7^ transformants, respectively. After being infected by the M13 helper phage, the phage displaying scFv antibodies were used for 5 rounds of bio-panning in ELISA plates which were immobilized with EV71-infected lysates (Figure 2A). In the 1st round, less than 1000 CFU were counted, however, the number increased with the rounds of bio-panning. This trend was observed in both short linker and long linker libraries. The amplified phages were further used to confirm their binding activities (Figure 2B). In the short linker library, the signals showed steadily positive signals against EV71-infected lysates during 1st to 5th round. In contrast, compared to the library, the long linker library showed a much more significantly increased signal during the 4th round, which also showed non-specific binding activities against BSA. The IgY antibody purified from the 7th immunization was included as a positive control. Taken together, these results suggested that specific clones might be enriched after 5 rounds of bio-panning.

### 2.3. Expression, Sequence Analysis, Classification, Binding Test, and Purification of scFv Antibodies

After the 5th round of bio-panning, the total DNA was purified and transformed into TOP10F’ competent *E. coli* for individual scFv antibody expression by isopropyl-beta-d-1-thiogalactopyranoside (IPTG) induction. A total of 26 clones were randomly selected from both short linker and long linker libraries to check their scFv expression in *E. coli*. Although no scFv antibodies were predominantly observed on SDS-PAGE, the expression of the recombinant scFv antibodies except clone EV71L3 were clearly detected as a 26 kDa protein band on Western blots (Figure 3A,B). Subsequent sequence analysis showed that a stop codon (UAG) was found in the EV71L3 of heavy chain, leading to the unsuccessful expression of EV71L3 scFv antibody. To determine the identity of the scFv antibody genes, the purified 26 scFv DNA from both short and long linker library were sequenced and characterized, respectively. The results revealed that all the scFv antibody clones from the short linker library have the same V_L_ and V_H_ gene usage as represented by EV71S1 (Table 1). Moreover, the sequence analysis indicated that 4 major groups using different gene usage were identified in the long linker library as represented by EV71L1 (15.38%), EV71L3 (7.69%), EV71L4 (69.23%), and EV71L9 (7.69%). The predicted amino acid of scFv antibodies were aligned with chicken germline for the determination of the mutation rate. The overall mutation rates of amino acids in the complementarity determining regions (CDRs) ranged from 0% to 89% in V_L_ regions and 40% to 71% in V_H_ regions (Table 2). In addition, the results also showed a significant mutation rate in the CDRs (0% to 89%) than in the framework regions (FRs) (0% to 25%), especially in the CDR3 regions (45% to 89%).

ELISA assays were performed to investigate their binding activities to the total lysates of EV71-infected cells. The IgY from the 7th immunization was used as a positive antibody control (Figure 3C). Only EV71S1 had specific binding activities whereas EV71L4 had a cross-reactive activity against BSA. No obvious binding activity was detected for EV71L1, EV71L3 and EV71L9. While the reason(s) for no binding activity of EV71L1 and EV71L9 was not known, it was ascribed to the stop codon mutation occurred in the V_H_ region of EV71L3 gene as mentioned above. EV71S1 scFv antibody which exhibited significant binding activity against the lysates of EV71-infected cells was purified and analyzed on the SDS-PAGE (Figure 4). The results indicated that the EV71S1 antibody was successfully purified as shown in the elution fraction (lane 3) while it was not totally eluted in the sepharose fraction (lane 4).

### 2.4. Cross-Reactivity of Chicken Derived IgY and scFv Antibodies

Moreover, we sought to test the cross-reactivity of the IgY and scFv antibodies against EV71-infected and *Coxsackievirus* 16 (CA16)-infected cell lysates. The ELISA assay showed that the IgY antibody was able to recognize both EV71-infected and CA16-infected lysates (Figure 5A). A similar result was observed when probed with EV71S1 scFv antibody. It was also observed that a significant binding signal was decreased in wells coated with non-infected lysates and BSA (Figure 5B). The Western blot was further applied for examining the cross-reactivity. The same preparation of cell lysates was separated and visualized on SDS-PAGE (Figure 5C). After being transferred to PVDF papers, a protein with a molecular weight of less than 37 kDa in EV71-infected lysates but not in CA16-infected and non-infected lysates was recognized by 7th-immunization IgY (Figure 5D). However, a protein with a molecular weight of more than 37 kDa was recognized by EV71S1 scFv antibody in all the cell lysates (Figure 5E). The results were contradictory to our expectations that they would be similar to the IgY results. The purity of virus particles used for chicken immunization might be a reason for the contradiction. But more additional experiments are needed, including identifying which proteins are recognized and using purer virus particles to figure out the discrepancy.

### 2.5. Neutralization Analysis of Chicken-Derived IgY and scFv Antibodies

The inhibition assay was performed to test the suppressive activities of EV71S1 scFv and IgY antibodies in reducing the cytopathic effect of viral infection. After virus infection to human rhabdomyosarcoma (RD) cells and incubation with antibodies, we calculated the half-maximal inhibitory concentration (IC_50_) for the determination of neutralization activity. The results showed that 7th-immunization IgY (Figure 6) and EV71S1 (Figure 6) possessed significant inhibitory activities, but pre-immunized IgY did not (Figure 6). The IC_50_ concentration of 7th-immunization IgY and EV71S1 were 0.045 ± 0.0019 µg/mL and 8.83 ± 0.05 µg/mL, respectively. These results implied that the antibody successfully inhibited the biological process of the virus infecting the RD cells.

## 3. Discussion

Virus infection including EV71 is a common public health problem that affects numerous lives globally. According to a report by the Centers for Disease Control and Prevention (CDC), EV71 and other enteroviruses cause more than 12 million symptomatic infections around the world each year [33]. EV71 infections have been associated with several outbreaks of HFMD across many countries [34]. As a result, there is a great need to develop a cost-effective and more tolerable therapeutic strategy against EV71. Immunotherapy using chicken antibodies might be a cost-effective and convenient candidate of strategy. For antibody generation, whether the process of antibody purification from animals is cost-effective or not is considered as an important aspect [35]. A small amount of antigens for chicken immunization can elicit a strong humoral immune response and with a relatively high titer that can persists for a longer period [36]. Unlike IgG antibodies of mammalian animals, chicken-derived IgY antibodies did not activate mammalian complement components or bind to Fc receptors, and rheumatoid factor [25]. Many studies have reported that using IgY as passive immunization might be a good substitute therapeutic approach [25,37]. In this study, we used the purified virus particles for chicken immunization to generate specific polyclonal IgY antibodies. By this method, the humoral immune responses were elicited (Figure 1). In addition, we demonstrated that these generated IgY antibodies had neutralization activities to EV71 in vitro assay (Figure 6). Although the usage of the IgY antibodies against EV71 virus in clinical is not feasible in a short time, our results offered an alternative way that the generation of antibodies with neutralizing activities in chicken is viable and more cost-effective, in particular when the underdeveloped countries and the immunogens are not readily available.

Monoclonal antibodies, including scFv antibody that recognizes a single antigenic epitope, have more potential in application to diagnostic and therapeutic agents. In contrast to traditional hybridoma technology needs isolation of B cells from the spleen, fusion of myeloma and B cells, separation of cell lines and screening of suitable cell lines, Huse et al. broke through the stereotypes and opened up new prospects for the generation of monoclonal antibody and the development of targeted therapies by constructing antibody libraries to select using phage display technology [38]. We constructed two libraries and selected scFv antibodies against EV71-infected cell lysate (Figure 2 and Figure 3). The short linker and long linker were used following the protocols previously described [26]. Additional linkers were also applied by other research groups. In our experiences, specific scFv antibodies with either the short linker or long linker could be selected through bio-panning [29]. The libraries constructed from hyper-immunized animals offered a more effective and feasible way to select specific antibodies compared to naïve libraries which needed more rounds of bio-panning to obtain specific clones [39]. In addition, it reduced the size of the library required to produce highly specific antibodies by using hyper-immunized animals as the source of immunoglobulin cDNA [40]. Otherwise, the integral procedure required for identifying specific antibodies was 1–2 months using phage display system, which is much quicker than traditional hybridoma technology requiring 3–5 months [41]. Taken together, considering both the expenditure and time, the phage display system is further confirmed to be superior to traditional hybridoma technology for generating specific antibodies [42,43]. However, the promiscuous pairing of V_L_ and V_H_ genes occurred in *E. coli* host may be disputable [42]. This problem could be answered by carrying out further experiments. Nevertheless, since the main purpose of our study was to obtain protein molecule(s) with binding activity, it may be more appropriate to name these *E. coli*-derived scFv molecules as “antibody-like molecules” or “recombinant binding proteins” to avoid such confusion.

During the bio-panning, we used the EV71 -infected cell lysate instead of EV71 virus particles because the virus particles were difficult to purify. In addition, we look forward to selecting antibody specificities that are closer to reality. As revealed by our bio-panning results, there was a gradual increase in eluted phage numbers until the third cycle. While after that, their numbers remained constant up to the fifth cycle (Figure 2A). On the other hand, the signals of amplified phage rose steadily in both libraries on ELISA (Figure 2B). However, in the long-linker library, the phages also recognized BSA, which means that these proteins might have the same epitope as EV71 virus-infected lysates or these phage expressed scFv is not specific. In the short linker library, we only selected one scFv antibody, EV71S1, after the bio-panning (Table 1). Based on our previous study, it is supposed that the only selected clone could be highly possible to be a dominant clone with high affinity [44]. The binding signal of EV71S1 was not as high as IgY, one reason might be the target protein has a low concentration in the lysates. Another reason might be that a panel of antibodies were elicited against different antigenic epitopes and presented in the IgY polyclonal antibodies, resulting in higher binding activities. In the long liker library, we did not select any specific scFv antibodies against EV71-infected cell lysate (Figure 3C). However, the randomly selected clones from the long linker library could still be classified into four groups and EV71L4 containing 69.23% was the dominant clone (Table 1). This indicated that the bio-panning procedure was performed successfully. We speculated that the form of scFv expressed in *E. coli* and on phage head, which might have a binding activity to EV71-infected cell lysate, had different conformation to affect the binding ability to EV71-infected cell lysate. In antibody, the variable V_L_ and V_H_ regions are generated via varied somatic mutation, V-J region recombination (additional D for V_H_), and miscellaneous heavy chains with light chains to create antibody diversity in most vertebrates [45,46]. The CDR regions of the antibodies are commonly considered to be the most divergent and important region against target proteins, especially the CDR3 region [47]. Therefore, the CDR usually had higher variation than germline after immunization. In this study, it is worth noting that mutation rates spanning from 19% to 64% were observed in total CDR, which was higher than that in the total FR of both V_L_ and V_H_ genes (Table 2). Notably, the mutation rates found in CDR3 regions of V_H_ fragments were much higher (45% to 89%) than in the CDR1 and CDR2 regions. Taken together, the results indicated that these clones with much greater variation in CDRs should be generated by an antigen driven immune response, rather than being directly from the naïve immunoglobulin repertoire. An additional limitation is that the stability, expression level, and purification efficiency of recombinant scFv are usually low and inconsistently from preparation to preparation as stated above.

The anti-EV71 IgY antibodies had cross-reactivity to CA16-infected lysate, but not to non-infected lysate and BSA on ELISA assay (Figure 5A). The EV71S1 also showed binding activities to both EV71-infected and CA16-infected cell lysate, and weakly cross-reacted to BSA (Figure 5B). These results were as anticipated because over 60% of the *enteroviruses* genome has been reported to be identical to that of *coxsackieviruses* [48]. A previous study also reported that human antibodies reacted against *enterovirus* and *coxsackieviruses* proteases when analyzed using ELISA [49]. On Western blot, the IgY specifically recognized only one band of approximately 30–37 kDa (Figure 5D). The reason is not exactly known; however, the result was highly reproducible. A similar phenomenon was observed when snake venom containing a mixture of proteins was used as an immunogen [29]. The result may be partially explained by the following descriptions: (i) IgY was used at a 1:5000 dilution on Western blot; (ii) the recognized protein might have more antigenicity in chicken, combined with the data in Figure 5D, it was reasoned that most IgY antibodies recognized this one protein. If the IgY was used at a higher concentration such as 1:1000 dilution, more bands should be recognized like in our previous studies. According to the previous studies, purified VPs of EV71 had anticipated molecular sizes of approximately 36 kDa, 32 kDa, 28 kDa, 27 kDa, and 8 kDa for VP0, VP1, VP2, VP3, and VP4, respectively [13]. As a fact, the IgY with neutralization activity might recognize VP0 or VP1. Previously, it was indicated that the VP4, which was produced by cleaving VP0, and VP1 both were played an important role in EV71 virus infection and replication [50,51]. This might be the reason for IgY to have the inhibitory effects for a decrease of EV71 infection in human RD cells. In contrast, the EV71S1 recognized not only the virus infected cell lysate but also the non-infected cell lysate, which protein was around 40–50 kDa on Western blot. It was reasoned that the purification of the virus particles used for chicken immunization still contained some cell proteins after using size exclusion column. After bio-panning by using infected cell lysate, the EV71S1 scFv antibody against cell proteins was selected. Worth noting is that EV71S1 still inhibited EV71 infection in human RD cells, even though the inhibition ability was lower than IgY antibodies (Figure 6C). We conjectured that EV71S1 might recognize a protein on RD cell which is important for EV71 infection, such as P-selectin glycoprotein ligand-1 (PSGL-1). PSGL-1, which molecular weight without transmembrane domain is approximately 45 kDa, is the receptor on cells for EV71 infection [52,53]. The EV71S1 might bind to the PSGL-1 to inhibit EV71 virus infection. However, further investigations are also required to determine viral active binding proteins, to rule out activity against host cell proteins and the sites of IgY and scFv antibodies that play a role in their inhibitory effects against EV71.

For the virus immunization, we referred to recent methods of developing human EV71 vaccines to have a better chicken immune response. Inactivation of the whole virus is the most common method to produce the viral vaccine compared to live-attenuated vaccines, VLP (virus-like particle)-based Vaccines, and recombinant VP1 and P1 Vaccines, due to more evidence and safety proving the effect of inactivated EV-A71 immunization is efficacy [54,55,56]. For the purification of the whole virus, we decide to use the size exclusion method, a 100 kDa spin column to separate viral proteins, instead of other conventional methods. The reasons are (1) Higher concentrations of the virus can be obtained in the case of limited animal immunization volume. (2) To accumulate a sufficient number of high-purity virus particles to have enough virus proteins such as VP1, VP2 or VP4 proteins. (3) It is simpler and faster to obtain viruses compared with traditional sucrose gradients [57,58]. We also tried first to obtain viral particles using a sucrose gradient, however, due to the unknown factors, the purity of the virus was not as good as expected. Therefore, there is no result to carry out the comparison on the effect of using these two purification methods for animal antibody generation. Thus, we can also see the bottleneck of the size exclusion method from the results of this study. It is speculated that the purification of this method may contain some cellular proteins, however, these proteins may also be closely related to the infection mechanism of the virus. This situation prevented us from producing monoclonal antibodies against the virus itself as originally planned, but allowed us to accidentally obtain antibodies with inhibitory activities on viral infection. We are aware that more experiments and evidence have to be done to prove the current hypothesis. The results obtained from this study can also assist readers to be more vigilant about the purification method, purity of antigens and whether there will be contamination of host cell proteins when conducting animal immunization for antibody production in the future. Although virus-specific monoclonal antibodies were not obtained after extensive screening, we believed that monoclonal antibodies against EV71 may be produced after immunization as evident by the inhibitory results of polyclonal antibodies. Unfortunately, we have not screened them at present, but we will continue to screen anti-EV71 monoclonal antibodies using the existing or newly constructed antibody libraries. Thus, we believe that there is a great potential to select monoclonal antibodies with high specificity and suppressive activities against EV71 viral proteins. Accordingly, as mentioned above, all of the IgY and scFv antibody libraries together would have great potential for the development of diagnostic reagents or therapeutic drugs for the prevention or therapeutic strategy of HFMD outbreaks in the future.

## 4. Materials and Methods

### 4.1. Animals and Ethics Declaration

All animal experiments were approved by the Institutional Animal Care and Use Committee of Taipei Medical University before project initiation. Female White Leghorn (*Gallus domesticus*) chicken at 6 months of age was used. Chickens were maintained in the Laboratory Animal Center of Taipei Medical University. (Ethical approval code: LAC-2015-0302, 28 March 2016).

### 4.2. Preparation of Virus-Infected Cell Lysate and Viral Particles

The Enterovirus 71 strain (EV71/4643/TW/98) was propagated in human rhabdomyosarcoma (RD) cells at Research Center for Emerging Viral Infections, Chang Gung University (Guishan Dist., Taoyuan City, Taiwan). The RD cells were grown in Dulbecco’s Modified Eagle Medium (DMEM) medium supplied with 10% of fetal bovine serum (Fisher Scientific, Fair Lawn, NJ, USA) at 37 °C with 5% CO_2_.

RD cells were infected with EV71 at a multiplicity of infection (MOI) of 10 and incubated for 6 h. After centrifugation, the infected cells were lysed in a buffer containing 50 mM Tris-HCl (pH 7.4), 150 mM sodium chloride (NaCl), 1% Triton X-100, and a protease inhibitor cocktail (Roche, Mannheim, Germany), inactivated by heating, and stored at −20 °C.

For the viral particle preparation, after infection and centrifugation, the viral particles present in the supernatant were inactivated by adding 37% formaldehyde (Merck, Søborg, Denmark) at a 1:4000 dilution, and incubating for 14 days at 37 °C. Then, the mixtures were loaded to the Amicon Ultra-4 Centrifugal Filter Unit 100 kDa (Merck Millipore, Darmstadt, Germany) for purification and concentration. The samples were stored at −80 °C.

### 4.3. Immunization of Chicken with EV71 Viral Particles

For the first immunization, 100 μg of EV71 viral particles were dissolved in PBS and mixed with an equal volume of complete Freund’s adjuvant (Sigma-Aldrich, Inc., St. Louis, MO, USA). Then it was administered into different areas of chicken thighs intramuscularly. For the subsequent immunizations, EV71 viral particles were mixed with an equal volume of incomplete Freund’s adjuvant (Sigma-Aldrich, Inc., St. Louis, MO, USA) and then administered at weekly intervals for seven times. The eggs were collected before the first immunization and after each immunization to purify total polyclonal IgY antibodies using dextran sulfate and sodium sulfate as described previously [59,60].

### 4.4. Construction of scFv Antibody Libraries

Phage displaying scFv antibody libraries were constructed as previously described [26]. In brief, chicken after 4 weeks of seventh immunization was sacrificed for extraction of total RNA from the spleens, which were homogenized in 5 mL TRIzol reagent based on the manufacturer’s guideline. Twenty micrograms of total RNA were used for cDNA synthesis by reverse transcriptase enzyme (Bionovas, Toronto, Canada). The cDNA was used for amplification of immunoglobulin light chain variable regions (V_L_) and heavy chain variable regions (V_H_) genes using specific primers, respectively. The PCR products (V_L_ and V_H_) were linked by flexible linkers (GQSSRSS or GQSSRSSSGGGSPGGGGS in amino acids) to make a full-length scFv antibody gene via overlapping PCR. After *SfiI* restriction enzyme (New England Biolabs [NEBs], Ipswich, CA, USA) digestion, the scFv genes were ligated into pComb3X and transformed into ER2738 (*E. coli*) by MicroPulser electroporation (Bio-Rad Micropulser, Hercules, CA, USA). A small amount of the transformed *E. coli* was plated on LB agar plates containing 50 µg/mL ampicillin (Amp) for determination of library size. The remaining bacteria were added to 100 mL of super broth containing 50 µg/mL Amp and then were infected with 10^11–^10^12^ PFU of VCS-M13 helper phage overnight. After centrifugation, the recombinant phages were precipitated on ice for at least 30 min using 4% (*w*/*v*) PEG-8000 and 3% NaCl. After another centrifugation, the precipitated phages were collected by resuspending in PBS containing 1% BSA and 20% glycerol, and then were stored at −20 °C.

### 4.5. Bio-Panning of scFv Antibodies

The selection of phage displaying scFv antibodies was performed in ELISA plates by bio-panning for five rounds. In short, EV71-infected lysates (100 µg/mL) were immobilized on the wells at 4 °C overnight. On the following day, the wells were blocked with 3% BSA in PBS at 37 °C for 1 h, and then the recombinant phages were added for incubation at 37 °C for 2 h. The ELISA wells were washed 10 times with each time containing 30 counts of vigorous pipetting using filtered PBST (PBS containing 0.05% Tween 20) to remove unbound phages. Bound phages were eluted from the wells using 0.1 M glycine-HCl (pH 2.2). After neutralizing by 2 M Tris base buffer, the eluted phages were amplified by infecting ER2738 *E. coli*, and a portion of infected cells was plated on LB agar plates containing 50 μg/mL of Amp for determination of eluted phage titers. The remaining infected cells were cultured overnight at 37 °C and collected the amplified phages for the next round of bio-panning.

### 4.6. Protein Expression and Purification of scFv Antibody

We purified total DNA from the 5th bio-panning and transformed it into TOP10F’ *E. coli* for the expression of individual scFv antibodies. The overnight culture of randomly selected clones was 1:100 diluted in the super broth medium containing 20 mM MgCl_2_ and 50 µg/mL of Amp. After culturing for 8 h at 37 °C, the 1 mM isopropyl-β-D-thiogalactopyranoside (IPTG) was added for protein induction. After centrifugation, the bacterial cells were resuspended in his-binding buffer (20 mM imidazole, 500 mM NaCl, 20 mM sodium phosphate, pH 7.4) and lysed by sonication. The scFv antibodies were purified using Ni^2+^ Sepharose according to the manufacturer’s protocol (GE Healthcare Bio-Sciences AB, Uppsala, Sweden).

### 4.7. Sequencing and Analysis of scFv Genes

The V_L_ and V_H_ genes of individual scFv DNA was sequenced and analyzed using primer ompseq (5′-AAGACAGCTATCGCGATTGCAGTG-3′) by the ABI 3730XL auto-sequencer machine (Applied Biosystems, Foster City, CA, USA). Their deduced amino acid sequences comprising frameworks (FRs) and complementarity determining regions (CDRs) were aligned and compared with that of the immunoglobulin germline of the chicken using the BioEdit alignment program [61].

### 4.8. SDS-PAGE and Western Blot

The EV71-infected lysates were separated on 15% SDS-PAGE under reducing conditions, which were transferred into PVDF membranes. After blocking with 5% skimmed milk in PBS at room temperature (25 °C) for 1 h, the scFv antibodies were detected using mouse anti-HA tag (Proteintech, Rosemont, IL, USA) and HRP-conjugated rabbit anti-mouse IgG (Jackson ImmunoResearch, West Grove, PA, USA) antibodies. The membranes were developed using the 3,3ʹ-diaminobenzidine (DAB) reagent. Similarly, the IgY was detected using HRP-conjugated donkey anti-chicken IgY (Jackson ImmunoResearch, West Grove, PA, USA). For the cross-reactivity assay, the selected scFv antibodies (10 μg/mL) were added and detected by mouse anti-HA tag and HRP-conjugated rabbit anti-mouse. Other steps, such as blocking, washing, incubation and color development steps were performed as described above.

### 4.9. Enzyme-Linked Immunosorbent Assay (ELISA)

The ELISA plates were coated with EV71-infected lysates (100 µg/mL) and incubated at 37 °C for 1 h. The wells were blocked with 5% skimmed milk in PBS for 1 h at 37 °C, and then purified IgY from pre-immunization and 7th-immunization were 2-fold serially diluted (500× to 256,000×) and added for incubation 1 h. After washing six times with PBST, the bound IgY antibodies were detected by HRP-conjugated donkey anti-chicken IgY. After six additional washings as above, the binding activities were detected by adding 3,3′,5,5′-tetramethylbenzidine (TMB). The reactions were stopped using 1 N HCl, and the optical density readings were taken at 450 nm. On phage-based ELISA, amplified phages containing 10^11^–10^12^ PFU from each round of bio-panning were used as probes and detected by HRP-conjugated mouse anti-M13 antibodies (GE Healthcare Bio-Sciences, Marlborough, MA, USA). To further confirm the specific binding activities, the selected scFv antibodies (10 μg/mL) were added to the ELISA wells immobilized with EV71 virus particles or EV71-infected lysates. Subsequently, mouse anti-HA and HRP-conjugated rabbit anti-mouse antibodies were used for detection. The blocking, washing, incubation, and color development were performed following the same conditions as described above. All of the ELISA data were represented as the mean ± SD from two independent experiments.

### 4.10. In Vitro Antibodies Neutralization Assay

The neutralization assay was used for measuring the ability of the antibody to inhibit the cytopathic effects induced by viruses. Briefly, 96-well tissue culture plates were seeded with 3 × 10^4^ cells/well in DMEM with 10% FBS and incubated at 37 °C for 24 h. The various concentration of antibodies were incubated with the virus at an MOI of 0.005 PFU/cell, which was the 100 TCID_50_s (a hundred time of 50% tissue culture infective doses), at 37 °C for 1 h [62]. The RD cells were infected by the mixture of antibodies and virus. After adsorption, the infected cells were covered with a medium containing 2% FBS, and further incubated at 37 °C for 64 h. The plates were fixed with 0.5% formaldehyde and then stained with 0.1% crystal violet. The density of the well at 570 nm was measured. Each experiment was performed in triplicate and repeated at least two times. The half-maximal inhibitory concentration (IC_50_) was calculated according to the formula IC_50_ = [(Y − B)/(A − B)] × (H − L) + L], where Y represents half of the mean optical density at 570 nm (OD570) of the cell control without the antibody, B represents the mean OD570 of wells with the antibody dilution nearest to and below Y, A represents the mean OD570 of wells with the antibody dilution nearest to and above Y, and L and H are the antibody concentrations at B and A, respectively.

### 4.11. Statistical Analysis

GraphPad Prism (version 6.0 software) was used for data analysis. The ELISA results were all performed duplicated, were analyzed and expressed as means plus or minus standard errors via multiple *t* tests. A *p* value of < 0.05 was considered statistically significant.

## 5. Conclusions

In conclusion, we carried out the immunization of EV71 virus with a cost-effective and convenient approach by simple virus purification using size exclusion and chicken immunization. The polyclonal IgY antibodies showed specific to EV71 virus and effectively inhibited the EV71 infection to RD cells. Additionally, we constructed scFv libraries and selected specific monoclonal antibody with phage display technology, which is more efficient in terms of cost and time for antibody generation. Unfortunately, the monoclonal antibody had no specific to EV71 virus. However, it still shows an inhibitory effect on EV71 virus infection, which led us to postulate that EV71S1 has the potential to recognize host cell proteins. Finally, reviewing the entire experimental design, we still believe that cost-effective and convenient approach is feasible, although no specific monoclonal antibody has been selected, but the polyclonal IgY antibodies results still have great potential for future development of diagnostic agents or treatments for EV71 infection. At the same time, scientists should also reflect about the purity of the virus when using animal post-immunization methods to generate antibodies to be more efficient.

## Figures and Tables

**Figure 1 ijms-23-12967-f001:**
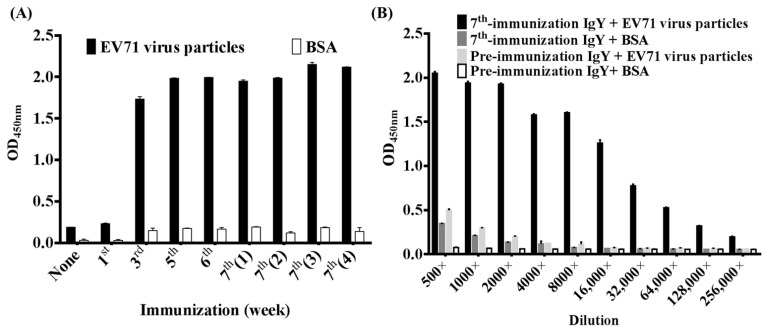
Humoral immune response in chicken. (**A**) The EV71 virus particles and BSA were immobilized on ELISA wells. Purified IgY from pre-immunization and each immunization were used to monitor the humoral immune response in chicken from the pre-immunized (None) to the fourth week after the 7th immunization (7th(4)). (**B**) Purified IgY antibodies from pre-immunization and the 7th immunization were in a series of two-fold dilutions (500× to 256,000×) to test the binding abilities to EV71 virus particles and BSA, respectively. ELISA experiments were performed in a duplicated manner and statistically analyzed as mean ± standard deviation (SD).

**Figure 2 ijms-23-12967-f002:**
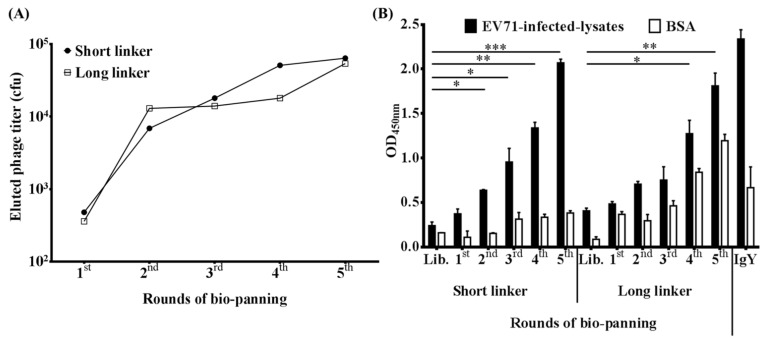
Determination of the phage titers and their specific binding activity after each round of bio-panning. (**A**) Two antibody libraries displaying scFv antibodies were constructed with short and long linkers and used for the bio-panning procedure. Recombinant phages were eluted to determine the colony-forming units (CFU) after each round of bio-panning. (**B**) Each amplified phage containing 10^11^ to 10^12^ plaque-forming units (PFU) was used as primary binders against EV71-infected lysates or BSA on phage-based ELISA. Purified IgY from the 7th immunization was used as a positive control. The labels Lib (original library phages) to 5th represent the phages isolated before and after 1 to 5 rounds of bio-panning. The experiments were performed in a duplicated manner and statistically analyzed as mean ± SD. *, *p* < 0.05; **, *p* < 0.01; ***, *p* < 0.001.

**Figure 3 ijms-23-12967-f003:**
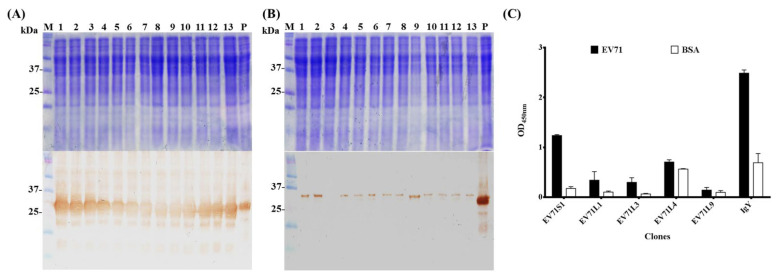
Expression and characterization of scFv antibodies. After bio-panning, 13 randomly selected clones from (**A**) short linker or (**B**) long linker library were induced by IPTG for scFv antibody expression. These expressed scFv antibodies were further confirmed using anti-HA antibody on Western blots. The expression of anti-*Coxsackievirus* 16 (CA16) S1 scFv antibody was included as a positive control (lane P). (**C**) After sequence analysis and classification, the binding activity of 5 represented clones, EV71S1, EV71L1, EV71L3, EL71L4 and EV71L9 scFv antibodies was tested against the lysate of EV71 infected cells at 1:10 dilution in ELISA. The IgY antibodies from 7th immunized chicken was used as a positive control. ELISA experiments were run in a duplicated manner and statistically analyzed as mean ± SD.

**Figure 4 ijms-23-12967-f004:**
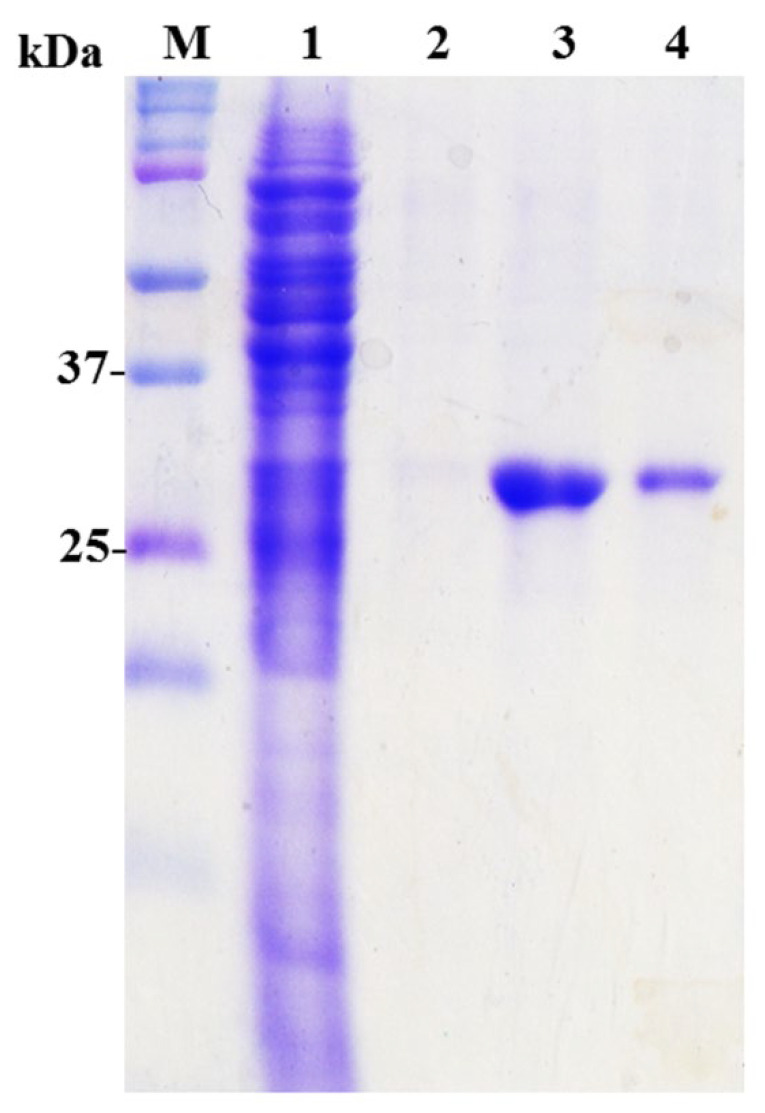
Purification of EV71S1 scFv antibody. After IPTG induction, the His-fused scFv was purified using Ni^2+^ sepharose and analyzed on SDS-PAGE stained with Coomassie blue dye. The purity of recombinant scFv antibody is genuinely high as demonstrated in lane 3. The scFv antibody was not entirely removed from the Ni^2+^ sepharose after elution (lane 4). Lane 1: flow through. Lane 2: washing. Lane 3: elution. Lane 4: sepharose.

**Figure 5 ijms-23-12967-f005:**
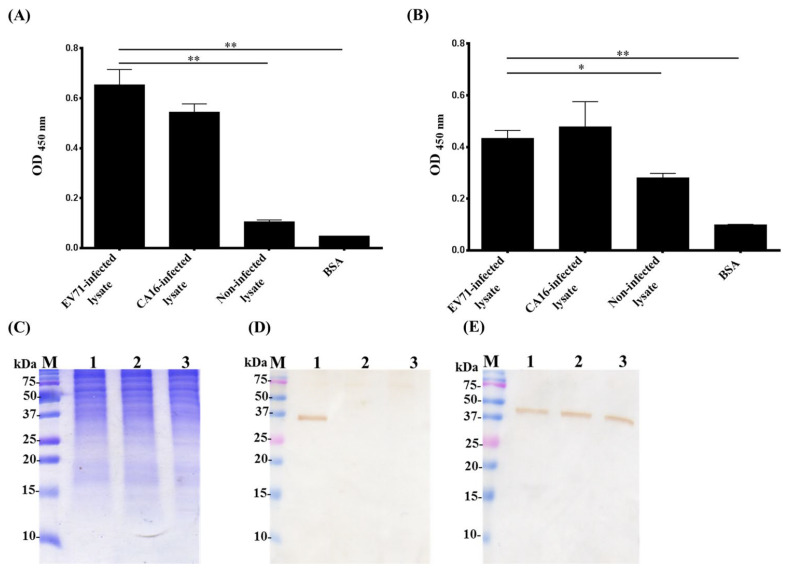
Cross-reactive analysis of IgY and EV71S1 antibodies against *Enteroviruses* 71. The EV71-infected, CA16-infected, non-infected cell lysates and BSA were coated on ELISA wells, respectively. The purified (**A**) IgY and (**B**) EV71S1 scFv antibodies were used for probing the viral proteins in lysates. *: *p* < 0.05. **: *p* < 0.01. (**C**) After SDS-PAGE separation, the PVDF membranes immobilized with EV71-infected (lane 1), CA16-infected (lane 2), and non-infected cell lysates (lane 3) were detected by purified (**D**) IgY and (**E**) EV71S1 scFv antibodies. ELISA experiments were run in a duplicated manner and statistically analyzed as mean ± SD.

**Figure 6 ijms-23-12967-f006:**
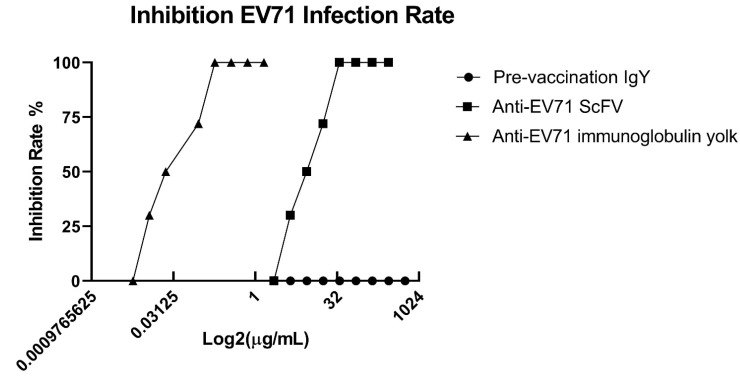
Analysis of IgY and scFv antibodies neutralizing activity. The EV71-infected RD cells were incubated with various concentrations of antibodies. After incubation, the density of the well at 570 nm was measured and the 50% inhibitory concentrations (IC_50_) were calculated according to the formula described in the material and method. All experiments were carried out in a triplicated manner.

**Table 1 ijms-23-12967-t001:** Grouping of chicken scFv clones according to the identities of V_L_ and V_H_ nucleotide sequences.

Group	Short Linker			Long Linker		
	V_L_	V_H_	Percentage	V_L_	V_H_	Percentage
Group 1	S1, S2, S3, S4 S5, S6, S7, S8, S9, S10, S11, S12, S13	S1, S2, S3, S4 S5, S6, S7, S8, S9, S10, S11, S12, S13	100%	L1, L2	L1, L2	15.38%
Group 2				L3	L3 *	7.69%
Group 3				L4, L5, L6, L7, L8, L10, L11, L12, L13	L4, L5, L6, L7, L8, L10, L11, L12, L13	69.23%
Group 4				L9	L9	7.69%

*: Containing stop codon in the open reading frame (ORF).

**Table 2 ijms-23-12967-t002:** Amino acid mutation rates of V_L_ and V_H_ genes used by scFv antibodies.

Clones	Region	FR1	FR2	FR3	FR4	Total FR	CDR1	CDR2	CDR3	Total CDR
EV71S1	*V_L_*	5%	6%	13%	0%	8%	22%	0%	45%	19%
	*V_H_*	7%	7%	16%	0%	9%	40%	47%	67%	52%
EV71L1	*V_L_*	5%	6%	9%	0%	6%	29%	14%	89%	48%
	*V_H_*	7%	21%	16%	0%	11%	60%	47%	71%	55%
EN71L4	*V_L_*	5%	13%	16%	0%	10%	38%	43%	63%	48%
	*V_H_*	7%	21%	16%	0%	11%	60%	47%	71%	55%
EN71L9	*V_L_*	5%	25%	9%	0%	10%	60%	43%	88%	64%
	*V_H_*	7%	21%	22%	9%	15%	60%	47%	71%	55%

CDRs: Complementarity determining regions; FRs: Framework regions; V_L_: variable domain of the light chain; V_H_: variable domain of the heavy chain.

## Data Availability

Not applicable.

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
