# Peer review of "The Bottlenecks of Preparing Virus Particles by Size Exclusion for Antibody Generation"

_ijms, 2022, doi:10.3390/ijms232112967_

Round 1

Reviewer 1 Report (New Reviewer)

Comments included in attached manuscript.

Author Response

We highly appreciate your valuable comments on our manuscript. We have revised our manuscript in accordance with the comments you suggested, and all the modifications made to the manuscript are in the red-colored font.

1. This sentence needs to be clarified. I believe the authors are trying to say that there are outbreaks of EV71 with a high mortality rate of 7% that have been reported in Europe.  I do not know what "early Europe" is referencing.

Thank you for your comment. We have removed the “early” in this sentence.

2. Antibody treatments are not considered an immunotherapy unless they are also used as an immunomodulator to direct the immune system against a specific pathogen. Consider changing the wording to antibody treatment.

Thank you for your kind suggestion. We have used “antibody treatment” instead as suggested.

3. This word needs to be changed as it does not seem to fit within the sentence. I think the authors are trying to indicate that the IgY antibodies are not strong inducers of the mammalian complement system but the meaning is not clear.

Thank you for your comment. We have revised this sentence for a better understanding.

4. It is not clear why the chickens were vaccinated weekly with EV71 virus particles rather than a more traditional strategy of vaccinating and then waiting for 2-4 weeks to develop and antibody response to the virus. Due to the method that was used, it is not clear whether the weekly vaccinations are necessary or if antibody responses would have appeared after a single administration of the EV71 virus particles.

Thank you for your question. In our previous studies, we have obtained strong antibody responses in the chickens administrated with other immunogens following the same vaccination protocol (Lee et al., 2016; Lee et al., 2017; Lee et al., 2018)

5. It does not appear that the authors completed any type of inactivation of the virus particles. Although not likely, it is possible the virus is replicating in the muscle tissue of the chicken which could be a concern for this type of an approach.

Thank you for your comment. The virus particles were inactivated by formaldehyde before the immunizations. We have provided the detailed information in section 4.2.

6. It is not clear what outcome the authors expected. Could the authors provide additional information on the potential problem and the proposed solution?

Thank you for your kind suggestion. We have modified the description with some elaborative information in section 2.4.

7. 50% inhibitory concentrations are not typically calculated for antibodies. Can the authors explain why they chose this method rather than the more standard approach of reporting the virus neutralization titer?

Thank you for your comment. We used 50% inhibitory concentrations in virion-based neutralization assays as described previously (Zheng, Zhu et al. 2019). These findings can suggest that the neutralizing antibodies neutralize via distinct mechanisms.

8. There does appear to be inhibition by the EV71S1 antibody but it is much lower than the inhibition provided by the 7th-immmunization IgY. Can the authors provide some explanation for the drastic decrease in neutralization?

Thank you for your question. Generally, a panel of antibodies with anti-EV71 activity were produced and presented in the IgY polyclonal antibodies. In addition to EV71S1, other scFv antibodies with similar or even better inhibitory activity t may not be selected from the constructed antibody libraries throughout the bio-panning procedure. This might be a reason why the IgY antibodies exhibited better inhibitory activity than EV71S1 scFv. We added the explanation to the 4th paragraph in the discussion.

9. This sentence is not clear. I am not sure what evidence is being presented that demonstrates that IgY antibodies are not feasible presently.

Thank you for your kind suggestion. We have modified the description in the first paragraph of discussion.

10. I would not expect a virus with 60% genome homology to be cross-reactive. It is an interesting finding, but definitely not what I would expect.

Thank you for your comment. We agreed with your observation. However, we have previously generated a monoclonal CaS1 scFv antibody from chickens immunized with enolase protein of C. albicans. This particular scFv antibody showed a strong binding activity against the enolase protein of S. aureus and S. pneumoniae with only 51-53% genome homology (Leu, Lee et al. 2020).

11. Why was such a high MOI used along with a short incubation period? This is highly unusual.

Thank you for your comment. We have consistently used high MOI with a short incubation period in enterovirus 71 infections to investigate the single viral life cycle and monitor the neutralizing mechanism (Lin, Weng et al. 2021).

12. The authors should explain their reasoning for this assay. For virus neutralization assays, antibody samples are typically mixed with virus containing media prior to addition to the cell culture monolayer.  What is the advantage to this method?

Thank you for your comment. We have mistakenly implanted the protocols to mimic the exact viral infection to its target cells in other studies. We did mix the antibodies with virus before adding to the cultured monolayer cells in neutralization assays as commonly performed. Accordingly, we have corrected the procedure in the section of Materials and Methods.

Reviewer 2 Report (New Reviewer)

The research described a production of polyclonal IgY in chicken and expression of scFv antibodies against EV-71 using phage display technology. The authors further characterized a binding and cross-reactivity of the generated molecules. It was primarily showed that the described molecules (IgY and scFc) may be beneficial in diagnosis or therapeutic, although more studies and characterizations are required. The methodology that the authors used could be useful for researchers who are interested in generation and characterization of antibodies against pathogen using phage display technology.

I have listed points that the authors could solve to improve the manuscript.

1.     Please carefully check a consistency of style that you used across the manuscript, i.e., the use of capital letters or small letters in title and reference (lines 115 and 136, lines 606, 609, etc.), the use of italic letter when mentioned organism name in Reference (lines 606, 666, etc.).

2.     Figure 1A – x-axis: Please clarify 7th (1), (2), (3), (4) in figure legend.

3.     Figure 1B – x-axis: Please check labels.

4.     Figure 2B – x-axis: What are 0, 1, 2 ,3 ,4 ,5 on the axis?

5.     Figure 5D: The IgY preparation is a polyclonal, why only one band was observed when reacting with virus-infected cell lysate? Please discuss.

6.     Figure 6: A more useful information could be added if the authors can provide a result of neutralization assay against CA16 virus to show the cross-reactivity in term of neutralization activity.

7.     Table 3 can be omitted. Its details can be read from the text.

8.     A discussion about using short linker and long linker should be provided. Why both short and long linkers were used in the study? What is the authors’ recommendation for selection of linker?

9.     Lines 299-301: Each established method has its pros and cons. To confirm that one method is superior than the other, more evidences or reference are needed for this statement.

10.  Lines 334-336: Could the authors classify more about this statement?

11.  Lines 354-357: The authors need to perform binding assays with purified virus or recombinant protein of EV71 or selectin to confirm their conjecture.

12.  Lines 369-371: What is a point that the authors would like to discuss about purified antigens, and pure virus particle?

13.  The authors should discuss the limitation of the study at the end of the discussion, and provide conclusion section.

14.  Materials and methods 4.2: How the virus-infected cell lysate is prepared, and how the virus particles are prepared? Please clarify. After lysing infected cells with lysis buffer, how could the authors be sure that intact virus particles were retrieved?

15.  Line 486: How did the authors get the virus in PFU unit? If the authors can titrate virus in PFU, why not the plaque reduction assay were used for neutralization test?

16. Could the authors clarify more about the neutralization assay used in this study or provide an original reference?

Author Response

We highly appreciate your valuable comments on our manuscript. We have revised our manuscript in accordance with the comments you suggested, and all the modifications made to the manuscript are in the red-colored font.

The research described a production of polyclonal IgY in chicken and expression of scFv antibodies against EV-71 using phage display technology. The authors further characterized a binding and cross-reactivity of the generated molecules. It was primarily showed that the described molecules (IgY and scFc) may be beneficial in diagnosis or therapeutic, although more studies and characterizations are required. The methodology that the authors used could be useful for researchers who are interested in generation and characterization of antibodies against pathogen using phage display technology.

Thank you for your friendly comments.

I have listed points that the authors could solve to improve the manuscript.

  1. Please carefully check a consistency of style that you used across the manuscript, i.e., the use of capital letters or small letters in title and reference (lines 115 and 136, lines 606, 609, etc.), the use of italic letter when mentioned organism name in Reference (lines 606, 666, etc.).

Thank you for your friendly reminder. However, it is the official regulation to have the italic letter in the sub-title and references.

  1. Figure 1A – x-axis: Please clarify 7th (1), (2), (3), (4) in figure legend.

Thank you for your kind comments. The same question was raised by another reviewer. Accordingly, we have clarified the figure legend.

  1. Figure 1B – x-axis: Please check labels.

Thank you for your kind suggestion. We have revised the labels in Figure 1B.

  1. Figure 2B – x-axis: What are 0, 1, 2 ,3 ,4 ,5 on the axis?

Thank you for your question. We have revised the labels in Figure 2B.

  1. Figure 5D: The IgY preparation is a polyclonal, why only one band was observed when reacting with virus-infected cell lysate? Please discuss.

Thank you for your comment. The reason is not exactly known. However, the result was highly reproducible. Similar phenomenon was observed when snake venom containing a mixture of proteins was used as an immunogen (Lee et al., 2016; Lee et al., 2017; Lee et al., 2018). We added the descriptions to the 4th paragraph in the discussion.

  1. Figure 6: A more useful information could be added if the authors can provide a result of neutralization assay against CA16 virus to show the cross-reactivity in term of neutralization activity.

Thank you for your comment. The stability, expression level and purification efficiency of recombinant EV71 scFv are usually low and inconsistently from preparation to preparation. It is difficult to accumulate enough amounts of testing scFv antibodies to carry out the same experiments against CA16 for comparison.

  1. Table 3 can be omitted. Its details can be read from the text.

Thank you for your kind suggestion. We have removed Table 3 as suggested.

  1. A discussion about using short linker and long linker should be provided. Why both short and long linkers were used in the study? What is the authors’ recommendation for selection of linker?

Thank you for your comments. The short linker and long linker were used following the protocols previously described (Andris-Widhopf, Rader, Steinberger, Fuller, & Barbas, 2000). Additional linkers were also applied by other research groups. In our experiences, specific scFv antibodies with either the short linker or long linker could be selected through bio-panning (Lee et al., 2016; Lee et al., 2017; Lee et al., 2018). We added the descriptions to the 2nd paragraph in the discussion.

  1. Lines 299-301: Each established method has its pros and cons. To confirm that one method is superior than the other, more evidences or reference are needed for this statement.

Thank you for your kind suggestion. We have added the references in 2nd paragraph of the discussion section.

  1. Lines 334-336: Could the authors classify more about this statement?

Thank you for your comment. We have revised the last sentence of 3rd paragraph in the discussion section.

  1. Lines 354-357: The authors need to perform binding assays with purified virus or recombinant protein of EV71 or selectin to confirm their conjecture.

Thank you for your comment. We are in the process of carrying out more experiments to reveal the identity of the protein recognized by EV71S1 antibody, including the generation of recombinant PSGL-1 protein or recombinant viral proteins.

  1. Lines 369-371: What is a point that the authors would like to discuss about purified antigens, and pure virus particle?

Thank you for your question. We included a detailed discussion relevant to the immunogen below (response to questions 13 and 14).

  1. The authors should discuss the limitation of the study at the end of the discussion, and provide conclusion section.

Thank you for your kind suggestions. We have tried to purify the virus particles using the sucrose gradient. However, the purity was not good as expected. Thus, we used the 100 kDa spin columns to separate the viral proteins from the cellular proteins for the subsequent chicken immunization. Thus, one of the major problems is to accumulate enough amounts of virus particles with high purity. Additional limitation is that the stability, expression level and purification efficiency of recombinant scFv are usually low and inconsistently from preparation to preparation as stated above. We have added more description in the last paragraph of the discussion and the “5. Conclusion” section as suggested.

  1. Materials and methods 4.2: How the virus-infected cell lysate is prepared, and how the virus particles are prepared? Please clarify. After lysing infected cells with lysis buffer, how could the authors be sure that intact virus particles were retrieved?

Thank you for your question. We have revised this section in detail. We used the infected cell lysates but not the intact viral particles for Western Blot assay.

  1. Line 486: How did the authors get the virus in PFU unit? If the authors can titrate virus in PFU, why not the plaque reduction assay were used for neutralization test?

We usually used high MOI (10) with a short incubation period in enterovirus 71 infections to investigate the single viral life cycle and monitor the neutralizing mechanism (Lin, Weng et al. 2021). As noted, the plaque reduction assay also can be used; however, the assay used in this study can better define the 50% inhibitory concentration of neutralizing antibodies in one experiment (Zheng, Zhu et al. 2019).

  1. Could the authors clarify more about the neutralization assay used in this study or provide an original reference?

Thank you for your question. The same question was also raised by another reviewer. We have mistakenly implanted the protocols to mimic the exact viral infection to its target cells in other studies. In this study, we did mix the antibodies with virus before adding to the cultured monolayer cells in neutralization assays as commonly performed. Accordingly, we have corrected the procedure in the section of Materials and Methods (Huang, Wang et al. 2020).

Reviewer 3 Report (New Reviewer)

Chi-Hsin Lee and colleagues conducted this investigation to show how chicken-derived EV71 antibodies may be used for EV71 infection detection and prevention. The isolated EV71 virus was used by the authors to immunize chickens and produce polyclonal yolk-immunoglobulin (IgY) antibodies. Additionally, using phage display technology, the authors created two single-chain variable fragment (scFv) antibody libraries. The findings showed that IgY and scFv antibodies exhibit distinct binding capabilities to Coxsackievirus A16 and EV71-infected cell lysates as well as an inhibitory effect on EV71 infection. Some information still needs to be clarified, particularly the antibodies' contradictory cross-reactivity.

Regarding the investigation of the antibodies' cross-reactivity, Fig. 5A and B's binding activity results by ELISA showed that both IgY and EV71S1 displayed cross-reactivity with CA16. Western blotting examination, however, reveals that the IgY only binds to the EV71-infected cell lysate and not the CA16-infected cell lysate (Fig. 5D). Could the authors share further information or an explanation of their reasoning?

By ELISA and Western blot, the EV71S1 antibody demonstrated cross-binding ability to the cell lysate of uninfected cells (Fig 5B and E). The antibody may also prevent EV71 infection of RD cells. The author hypothesized that the antibody's recognition of the cellular PSGL-1 protein may be the cause of the cross-binding to non-infected cells and the infection-inhibitory impact. Would the author be able to provide a more detailed explanation of why the monoclonal antibody created by immunizing with the purified viral particle has a significant binding to cellular protein but not viral protein? To address the issue of the antibodies binding to viral or host proteins, have the authors performed ELISA or Western blot utilizing the EV71 and CA16 viral particles instead of the infected cell lysate?

The authors state on line 318-9 that the lower binding signal of EV71S1 compared to IgY is due to the low concentration of the target protein in the lysates. The author may consider that the multiple binding epitopes of the polyclonal antibody have an effect on this higher binding activity.

Minor adjustments:

1. Please explain why the vaccination was necessary seven times.

2. The author should provide statistical analysis data in all relevant figures, particularly the results for which the author stated a significant difference (line 150-1). Also, in the materials and methods section, please provide more information about the statistical analysis method.

3. Figure 1A: Could you please explain the numbers in brackets? (after 7th)

4. Figure 1B, change ? to X

5. Please keep the graph pattern consistent throughout (such as font size and scales)

Author Response

We highly appreciate your valuable comments on our manuscript. We have revised our manuscript in accordance with the comments you suggested, and all the modifications made to the manuscript are in the red-colored font.

Chi-Hsin Lee and colleagues conducted this investigation to show how chicken-derived EV71 antibodies may be used for EV71 infection detection and prevention. The isolated EV71 virus was used by the authors to immunize chickens and produce polyclonal yolk-immunoglobulin (IgY) antibodies. Additionally, using phage display technology, the authors created two single-chain variable fragment (scFv) antibody libraries. The findings showed that IgY and scFv antibodies exhibit distinct binding capabilities to Coxsackievirus A16 and EV71-infected cell lysates as well as an inhibitory effect on EV71 infection. Some information still needs to be clarified, particularly the antibodies' contradictory cross-reactivity.

Regarding the investigation of the antibodies' cross-reactivity, Fig. 5A and B's binding activity results by ELISA showed that both IgY and EV71S1 displayed cross-reactivity with CA16. Western blotting examination, however, reveals that the IgY only binds to the EV71-infected cell lysate and not the CA16-infected cell lysate (Fig. 5D). Could the authors share further information or an explanation of their reasoning?

Thank you for your comment. The reason is not exactly known. However, we have occasionally obtained antibodies recognizing the target antigen coated on ELISA plates but not on Western blots, which may result from the different conformation of the same antigen presented in two methods (Lee, Lee et al. 2017).  

By ELISA and Western blot, the EV71S1 antibody demonstrated cross-binding ability to the cell lysate of uninfected cells (Fig 5B and E). The antibody may also prevent EV71 infection of RD cells. The author hypothesized that the antibody's recognition of the cellular PSGL-1 protein may be the cause of the cross-binding to non-infected cells and the infection-inhibitory impact. Would the author be able to provide a more detailed explanation of why the monoclonal antibody created by immunizing with the purified viral particle has a significant binding to cellular protein but not viral protein? To address the issue of the antibodies binding to viral or host proteins, have the authors performed ELISA or Western blot utilizing the EV71 and CA16 viral particles instead of the infected cell lysate?

Thank you for your question. We have tried to purify the virus particles using the sucrose gradient. However, the purity was not good as expected. Thus, we used the 100 kDa spin columns to separate the viral proteins from the cellular proteins for the subsequent chicken immunization. It might be a reason the speculated anti-PSGL-1 scFv antibody was selected by bio-panning. We did not carry out the experiments as suggested due to the difficulties to accumulate enough amounts of purified virus particles.

The authors state on line 318-9 that the lower binding signal of EV71S1 compared to IgY is due to the low concentration of the target protein in the lysates. The author may consider that the multiple binding epitopes of the polyclonal antibody have an effect on this higher binding activity.

Thank you for your comment. We are aware that a panel of antibodies were elicited against different antigenic epitopes and presented in the IgY polyclonal antibodies.  We added the explanation to the 4th paragraph in the discussion.

Minor adjustments:

  1. Please explain why the vaccination was necessary seven times.

Thank you for your question. We immunized the chicken for seven times to assure the maximal antibody response could be obtained based on our previous studies (Lee et al., 2016; Lee et al., 2017; Lee et al., 2018).

  1. The author should provide statistical analysis data in all relevant figures, particularly the results for which the author stated a significant difference (line 150-1). Also, in the materials and methods section, please provide more information about the statistical analysis method.

Thank you for your comment. We have revised the sentence, figure legend in Figure 2, Figure 2B, and the section of Statistical Analysis.

  1. Figure 1A: Could you please explain the numbers in brackets? (after 7th)

Thank you for your question. We have clarified the figure legend in Figure 1A.

  1. Figure 1B, change ? to X

 Thank you for your kind comment. We have made the change in Figure 1B as your suggestion.

  1. Please keep the graph pattern consistent throughout (such as font size and scales)

 Thank you for your kind comments. We have tried to make the revisions based on your suggestions as much as possible.

Round 2

Reviewer 2 Report (New Reviewer)

The authors have revised the manuscript accordingly. There are only some minor points.

Fig 2B - Please define ‘Lib.’

Line 371 – check typing

Lines 515-516 – [= 100 × 50% tissue culture infective doses (TCID50s)], please clarify this phrase. It makes confusion. 

Author Response

Fig 2B - Please define ‘Lib.’

Thank you for your kind suggestion. We have defined the term as requested.

Line 371 – check typing

Thank you for your kind comment. We have made the correction.

Lines 515-516 – [= 100 × 50% tissue culture infective doses (TCID50s)], please clarify this phrase. It makes confusion. 

Thank you for your comment. We have clarified the sentence and added the reference.

Reviewer 3 Report (New Reviewer)

The author attempted to revise the manuscript in response to the reviewers' comments; however, the quality of figure preparation could be improved, and revised texts should be carefully checked.

Figure 2: The pattern is inconsistent, for example. The term "bio-panning" or "biopanning," the axis labeling font size, and the superscripts

Figure 6: The new figure has been replaced, but the author has not revised the figure legend, which includes the relevant result section (subheading 2.5, lines 244-253) and relevant discussion section (line 368)

Line 371; Please check and correct inhibitory activity t may not be selected

Line 516; Please check and correct TCID50s

Line 673; Reference no.43 seems not formatted. 

Line 705; Reference no.57 format is not consistent.

Author Response

Figure 2: The pattern is inconsistent, for example. The term "bio-panning" or "biopanning," the axis labeling font size, and the superscripts

Thank you for your comment. We have modified the terms in a consistent way.

Figure 6: The new figure has been replaced, but the author has not revised the figure legend, which includes the relevant result section (subheading 2.5, lines 244-253) and relevant discussion section (line 368)

Thank you for your comment. We have revised the figure legend. The x-axis shown in log2 scale was recommended by another reviewer. However, the results and descriptions remained the same.

Line 371; Please check and correct inhibitory activity t may not be selected

Thank you for your comment. We have made the correction.

Line 516; Please check and correct TCID50s

Thank you for your comment. We have revised the sentence and added the reference.

Line 673; Reference no.43 seems not formatted. 

Thank you for your comment. We have made the correction.

Line 705; Reference no.57 format is not consistent.

Thank you for your comment. We followed the format requested by the Journal. However, the bracket appeared automatically in the title of reference #57 when EndNote was applied.

This manuscript is a resubmission of an earlier submission. The following is a list of the peer review reports and author responses from that submission.

Round 1

Reviewer 1 Report

please find necessary comments need to be placed before publication

Author Response

this statement does not deliver any scientific output. I would suggest replacing with a conclusion statement which indicates scientific take aways. Also a future recommendation sentence after that would be helpful.

  1. Thank you for the kind suggestion. We have made the sentence more conclusive to show the scientific takeaway.

its fine if it is reported in different regions. but what is interesting in these. Any numbers? any level of severity or anything numeric can help to show importance of it?

  1. We included more detailed information with the mortality rate percentage to help the readers to have a better understanding.

this could be in methods section

  1. Thank you for your appropriate suggestion. Thus, we deleted the description here since a similar description has been given in the methods section.

hypothesis could be more clear.

  1. Thank you for the kind suggestion. Accordingly, we have rephrased the ending of the introduction and clarified the hypothesis.

quality of figure is low must be improved

  1. Thank you for the kind suggestion. We have replaced Figure 1 with higher quality.

same here

  1. We have replaced Figure 2 with higher quality.

same quality must be improved. Also what error bars are those? standard or percentage?

  1. We have replaced Figure 3 with higher quality. The error bars in figure 3 (B) represented the statistical analysis of duplicated experimental results as mean ± SD. The results are the original numerical values from the optical density at 450 nm.

no need to highlight rows on tables. I cant see how the statistical differences are placed here?

  1. Thank you for your kind suggestion. Accordingly, we have removed the highlight parts.

same as table 1

  1. Thank you for your kind suggestion. Accordingly, we have removed the highlight parts.

same

  1. Thank you for the kind suggestion. We have replaced Figure 5 with higher quality.

this should go before results section I think

  1. Thank you for your kind suggestion. However, it is the official regulation to have the methods section after the results section by the journal.

well this section need to be strengthened with more details. entire paper is in very detailed but not this part. this part should be better explained compare to other sections.

  1. Thank you for your kind suggestion. Accordingly, we have largely modified this section by providing more detailed information.

Reviewer 2 Report

Lee and coworkers here describe the generation of chicken IgY antibodies and single chain variable fragments against EV-A71.  After 7 immunizations of chickens with their EV-A71 viral particles, the authors pool IgY and use splenic RNA samples to clone the variable regions of the B cells that were induced to make antibody.  After initial confirmation that the IgY was EV-A71-specific in ELISA, the single chain variable fragments that were cloned into  phage display libraries were tested for binding.  After multiple rounds of biopanning, the short and long linker scFv molecule phage pools showed reactivity to EV-A71 antigen in ELISA.  Individual scFv clones had some reactivity to the EV-A71 antigens used in ELISA but also seemed to have cross-reactivity with uninfected cell lysates in ELISA and also on western blot.  The IgY pool showed very strong neutralization of EV-A71 in cell culture inhibition assays but the selected scFv (EV71S1) had 200-fold lower neutralization capability.  Although the authors discuss the possibility that the EV71S1 scFv might actually be specific for a rhabdomyosarcoma cell protein, the authors conclude that the data presented suggests that IgY and scFv molecules have a great potential for future EV-A71 therapeutic treatments. 

Overall critique:  The IgY antibody pool likely contains some very potent neutralizing antibodies against EV-A71.  As the stated purpose of this manuscript was the generation and evaluation of chicken antibodies against EV-A71 using phage display, it seems that the scFv molecule that was down selected from the phage display library is neither specific for nor a potent neutralizing antibody against EV-A71.  Thus, the use of phage display and scFv technology failed, strongly suggesting that the authors' conclusions do not follow from the data.  It is altogether possible that potent neutralizing scFv molecules could be pulled out of the phage display libraries generated here and this would support the authors' conclusions.  Until this is done, this manuscript should not be published.

Specific critiques: 

1) There are many possible reasons for the failure to generate EV-A71 specific scFv molecules.  The most likely one is that the viral particles that were used to immunize the chickens was not pure.  Based on the materials and methods, it seems that the virions were prepared by lysing infected cells, clarifying the lysate then centrifuging it on 100 kDa spin columns.  This definitely enriched for viral particles, but it also enriched for all of the soluble host proteins that are greater than 100 kDa or that could not fit through the pores of the column.  It is very likely that those non-specific RD cell proteins are greater than 90% of all of the protein in the viral particle preparation.  Please run your particle preparation on an SDS/PAGE gel and stain with coomassie - this will likely demonstrate that most of the protein is not EV-A71.  You then will need to go back and generate sucrose density gradient-purified virions or use some sort of FPLC purification to make their viral particles for immunization.  This will greatly increase the chances of eliciting a higher proportion of EV-A71 specific B cells for the spleen RNA extraction.

2) There are many statements in the manuscript's introduction that are misleading.  Picornaviruses all have the same capsid structure and is a very large family of viruses.  Statements on line 54 and 68 seem to imply that picornaviruses are only made up of human enteroviruses and coxsacieviruses.  There is a reference on line 98 that is used to back up the statement that passive immunization of neutralizing proteins could provide protection in mice from fatal EVA71 infection in vivo.  This manuscript describes antibody generation presumably for passive transfer into infected children.  The referenced manuscript described the immunization of female mice passing on maternal antibody to their babies. Perhaps a manuscript that describes passive transfer of antibodies elicited from EV-A71 vaccination or infection would be more suitable here.

3) In many of the ELISA graphs (2B, 3C, 5A, 5B), there are comparisons of different antibodies against infected cell lysates or uninfected cell lysates.  Statements are made suggesting some of the binding is "low" or "not obvious".  But comparing EV71L1 in figure 3C ("not binding")and EV71S1 in figure 5B ("specific binding"), the OD's for both are very close to 0.4.  Can you draw a line on each of these graphs that demonstrates the binding/not binding OD cutoff?  

4) Figure 6 needs to be 1 graph, where A, B and C are all plotted on the same graph with the same x axis on a log 2 scale.  Line 243 suggests the inhibitory activity of scFv is "significant" yet it is 200-fold lower than the IgY pool and in general, EV-A71 antibodies that have neutralizing activities above 1 µg/ml are not considered "good" or "strong".

5) Line 280 in the discussion mentions "stereotypes".  Are you trying to say that scientists were resistant from moving away from hybridomas?  At this point, technology is moving past phage display and cloning rearranged antibody RNAs to directly amplifying, sequencing and producing intact antibodies from spleen RNA sequences, and these can be produced in less than a week, spleen to antibody-containing supernatants, no bacteria required.  Statements about phage display's flexibility and speed on lines 291/292 will not age well.

6) The statement on line 305/306 about EV-A71 and BSA having similar epitopes needs to be removed.  The more likely explanation is that the scFv is not specific for EV-A71.

7) Most of the discussion is spent explaining what may have gone wrong with the scFv production.  In general, a discussion should explain how the work presented in the results section fits in with the greater body of existing knowledge on the topic.  Have any other labs generated strongly neutralizing scFv that originated with a partially purified virus vaccine for any virus?  Have scFv been successfully used to prevent disease in animal models for picornaviruses or any other virus?

Author Response

Lee and coworkers here describe the generation of chicken IgY antibodies and single chain variable fragments against EV-A71.  After 7 immunizations of chickens with their EV-A71 viral particles, the authors pool IgY and use splenic RNA samples to clone the variable regions of the B cells that were induced to make antibody.  After initial confirmation that the IgY was EV-A71-specific in ELISA, the single chain variable fragments that were cloned into  phage display libraries were tested for binding.  After multiple rounds of biopanning, the short and long linker scFv molecule phage pools showed reactivity to EV-A71 antigen in ELISA.  Individual scFv clones had some reactivity to the EV-A71 antigens used in ELISA but also seemed to have cross-reactivity with uninfected cell lysates in ELISA and also on western blot.  The IgY pool showed very strong neutralization of EV-A71 in cell culture inhibition assays but the selected scFv (EV71S1) had 200-fold lower neutralization capability.  Although the authors discuss the possibility that the EV71S1 scFv might actually be specific for a rhabdomyosarcoma cell protein, the authors conclude that the data presented suggests that IgY and scFv molecules have a great potential for future EV-A71 therapeutic treatments. 

Overall critique:  The IgY antibody pool likely contains some very potent neutralizing antibodies against EV-A71.  As the stated purpose of this manuscript was the generation and evaluation of chicken antibodies against EV-A71 using phage display, it seems that the scFv molecule that was down selected from the phage display library is neither specific for nor a potent neutralizing antibody against EV-A71.  Thus, the use of phage display and scFv technology failed, strongly suggesting that the authors' conclusions do not follow from the data.  It is altogether possible that potent neutralizing scFv molecules could be pulled out of the phage display libraries generated here and this would support the authors' conclusions.  Until this is done, this manuscript should not be published.

Thank you for your kind comments. As noted, the polyclonal IgY antibodies showed significant inhibitory effect on EV71-infected RD cells, indicating that scFv antibodies with neutralization activity present in the constructed antibody libraries. Although the selected EV71S1 antibody recognized a non-specific protein on western blot (figure 5E), it showed significant inhibitory ability on virus-infected RD cells (Figure 6C). Even though this is a discrepant observation firstly, it could be reasoned that EV71S1 antibody in deed recognized a cellular component playing an important role in EV71 infection. Presently, we do not have any solid evidence to support our speculation. We are in the process of carrying out more experiments to reveal the identity of the protein recognized by EV71S1 antibody.

Specific critiques: 

1) There are many possible reasons for the failure to generate EV-A71 specific scFv molecules. The most likely one is that the viral particles that were used to immunize the chickens was not pure.  Based on the materials and methods, it seems that the virions were prepared by lysing infected cells, clarifying the lysate then centrifuging it on 100 kDa spin columns.  This definitely enriched for viral particles, but it also enriched for all of the soluble host proteins that are greater than 100 kDa or that could not fit through the pores of the column.  It is very likely that those non-specific RD cell proteins are greater than 90% of all of the protein in the viral particle preparation.  Please run your particle preparation on an SDS/PAGE gel and stain with coomassie - this will likely demonstrate that most of the protein is not EV-A71.  You then will need to go back and generate sucrose density gradient-purified virions or use some sort of FPLC purification to make their viral particles for immunization.  This will greatly increase the chances of eliciting a higher proportion of EV-A71 specific B cells for the spleen RNA extraction.

Thank you for your kind comments. We totally agree that the purity of the immunogen is very low. We have ever tried to purify the virus particles using the sucrose gradient as you mentioned. However, the quality was not good as expected. Moreover, the total amount of virus particles was not enough for eliciting strong humoral antibody response. Thus, that is the major reason that we used the 100 kDa spin columns to separate the virus particles from the most cellular proteins for the subsequent immunization.

2) There are many statements in the manuscript's introduction that are misleading.  Picornaviruses all have the same capsid structure and is a very large family of viruses.  Statements on line 54 and 68 seem to imply that picornaviruses are only made up of human enteroviruses and coxsacieviruses.  There is a reference on line 98 that is used to back up the statement that passive immunization of neutralizing proteins could provide protection in mice from fatal EVA71 infection in vivo.  This manuscript describes antibody generation presumably for passive transfer into infected children.  The referenced manuscript described the immunization of female mice passing on maternal antibody to their babies. Perhaps a manuscript that describes passive transfer of antibodies elicited from EV-A71 vaccination or infection would be more suitable here.

Thank you for your kind comments. Accordingly, we revised the statements and the references.

3) In many of the ELISA graphs (2B, 3C, 5A, 5B), there are comparisons of different antibodies against infected cell lysates or uninfected cell lysates.  Statements are made suggesting some of the binding is "low" or "not obvious".  But comparing EV71L1 in figure 3C ("not binding")and EV71S1 in figure 5B ("specific binding"), the OD's for both are very close to 0.4.  Can you draw a line on each of these graphs that demonstrates the binding/not binding OD cutoff?  

Thank you for your kind comments. These ELISA experiments were not done at the same time. We analyzed these results by comparing them with the BSA as a negative control. The expression level and purification efficiency of recombinant EV71 scFv is usually low and various from preparations to preparations. Due to this season, we used the cellular lysates containing scFv antibody to test the binding activity (Figure 3C), and the purified scFv antibody (10 μg/mL) to test the cross-reactivity (Figure 5B). Thus, the concentration of EV71S1 applied was different and not comparable in the experiments performed in Figure 3C and Figure 5B.

4) Figure 6 needs to be 1 graph, where A, B and C are all plotted on the same graph with the same x axis on a log 2 scale.  Line 243 suggests the inhibitory activity of scFv is "significant" yet it is 200-fold lower than the IgY pool and in general, EV-A71 antibodies that have neutralizing activities above 1 µg/ml are not considered "good" or "strong".

Thank you for your kind comments. We revised Figure 6 as suggested. It is well known that a panel of antibodies with anti-EV71 activity is included in the IgY polyclonal antibodies. However, the scFv antibodies with high inhibitory activity than EV71S1 scFv may not be selected from the constructed antibody libraries throughout the bio-panning procedure. This might be a reason why the IgY antibodies exhibited better inhibitory activity than EV71S1 scFv. We are in the process of generating the scFv-Fc fusion antibody to improve its anti-EV71 activity.

5) Line 280 in the discussion mentions "stereotypes".  Are you trying to say that scientists were resistant from moving away from hybridomas?  At this point, technology is moving past phage display and cloning rearranged antibody RNAs to directly amplifying, sequencing and producing intact antibodies from spleen RNA sequences, and these can be produced in less than a week, spleen to antibody-containing supernatants, no bacteria required.  Statements about phage display's flexibility and speed on lines 291/292 will not age well.

Thank you for your kind comments and letting us know the newly developed technology for making monoclonal antibodies. In present, hydridoma and phage display technology are the most available and applied methods in generating specific monoclonal antibodies. Here, we simply intended to describe and compare the merits of these 2 novel technologies. Thus, we modified the description as well.

6) The statement on line 305/306 about EV-A71 and BSA having similar epitopes needs to be removed.  The more likely explanation is that the scFv is not specific for EV-A71.

Thank you for your kind comments. Accordingly, we revised this statement.

7) Most of the discussion is spent explaining what may have gone wrong with the scFv production.  In general, a discussion should explain how the work presented in the results section fits in with the greater body of existing knowledge on the topic.  Have any other labs generated strongly neutralizing scFv that originated with a partially purified virus vaccine for any virus?  Have scFv been successfully used to prevent disease in animal models for picornaviruses or any other virus?

Thank you for your kind comments. Accordingly, we have revised the last paragraph in the discussion section, providing more detailed information on scFv antibodies against other picornaviruses.

Round 2

Reviewer 2 Report

In the rebuttal to my critique, the authors agree that their Virion preparation to be used as antigen to elicit chicken antibodies was not pure and resulted in poor yields.  Thus, the premise of this manuscript, that chicken antibodies can be elicited from EVA71 vaccination and then panned/downsampled to identify strongly neutralizing antibodies is not supported.  The authors did not change anything related to this fundamental flaw, I suggest at minimum that if the authors would like to resubmit this manuscript they return to the phage display library and pan until they find an EVA71-specific clone.  Perhaps a better course of action would be to purchase EVA71 purified particles or collaborate with a laboratory that can generate these particles for them, using that antigen to elicit a more EVA71-specific set of IgY.

The idea that chicken vaccination with an extremely impure Virion preparation could elicit strongly neutralizing monoclonal antibodies is not out of the realm of possibility, but the data in this manuscript does not support this hypothesis.

Alternatively, a new manuscript could be presented where the chicken antibodies are tested in an in vivo challenge system to demonstrate efficacy.  But there might be concerns that antibodies against RD cell proteins might cause cross-reactive problems in the animal host.